# LIPEx – Locally Interpretable Probabilistic Explanations – To Look Beyond The True Class

## Abstract

In this work, we instantiate a novel perturbation-based multi-class explanation framework, LIPEx (**L**ocally **I**nterpretable **P**robabilistic **Ex**planation). We demonstrate that LIPEx not only locally replicates the probability distributions output by the widely used complex classification models but also provides insight into how every feature deemed to be important affects the prediction probability for each of the possible classes. We achieve this by defining the explanation as a matrix obtained via regression with respect to the Hellinger distance in the space of probability distributions. Ablation tests on text and image data, show that LIPEx-guided removal of important features from the data causes more change in predictions for the underlying model than similar tests on other saliency-based or feature importance-based XAI methods. It is also shown that compared to LIME, LIPEx is much more data efficient in terms of the number of perturbations needed for reliable evaluation of the explanation.

## 1 Introduction

Recent momentum in deep learning research has made interpreting models with complex architectures very important. In a wide range of areas where neural nets have made a successful foray, the method of "Explainable A.I." (XAI) has also found an important use to help understand the functioning of these novel predictors - like in climate science (Labe & Barnes, 2021), for solving partial differential equation (Linial et al., 2023), in high-energy physics (Neubauer & Roy, 2022), information retrieval (Lyu & Anand, 2023), in legal A.I. (Collenette et al., 2023), etc. Most often, it has been observed that models with complex architectures give better accuracy compared to a simple model. So, the core puzzle that XAI can be seen to solve is to give a highly accurate local replication of a complex predictor's behaviour by a simple model over humanly interpretable components of the data (Ribeiro et al., 2016). Towards achieving this, multiple different XAI methods have been proposed in the recent times, e.g., LIME (Ribeiro et al., 2016), SHAP (Lundberg & Lee, 2017), Decision-Set (Lakkaraju et al., 2016), Anchor (Ribeiro et al., 2018), Smooth-GRAD (Smilkov et al., 2017b), Poly-CAM (Englebert et al., 2022), Extremal Perturbations (Fong et al., 2019), Saliency Maps (Simonyan et al., 2014), etc.

One major motivation for explainability is debugging a model (Casillas et al., 2003; Dapaah & Grabowski, 2016). Towards this, an end user is interested not only in understanding the explanation provided for the predicted class at a particular data point but also in the influence of different features for all possible class likelihoods estimated by a classifier. The full spectrum of feature influence on each class at a particular data point can help to understand how well the model has been trained to discriminate a particular class from the rest. However, existing explanation frameworks do not provide any clue on the aforementioned issue. To this end, we propose an explainability framework that can explain a classifier's output prediction beyond the true class.

To obtain an explanation around a data point, a local explanation algorithm like LIME (Garreau & Luxburg, 2020) creates perturbations around it, each perturbation being represented as a Boolean vector. LIME includes a feature selection method to decide a set of important features for each class (like Algorithm A) among which the perturbations are considered. Then, an explanation vector for the complex model's prediction on the input data is obtained by solving a penalized linear regression over these perturbations and the complex classifier's predictions on the data corresponding to the perturbations. We posit that it is not entirely convincing that LIME attempts to regress over bounded labels, i.e., probabilities, using an unbounded function (i.e., a linear function) and that this would need to be called separately for each class. Further, even if by repeated calls on each possible class

we obtain an explanation for of the classes, there is no guarantee that by these repeated evaluations, the importance of any particular feature would be knowable for every class.

In this work, we attempt to remedy these problems by proposing a single unified framework that applies to both text and images, which we will show in experiments to be better than various XAI methods for both text and images. In a $\mathcal{C}$−class classification task, for any data $s$ which is represented as $\mathbf{z}_s$ in some $f_s$ dimensional feature space, we shall seek explanations that map into $\mathcal{C}$−class probability space as,

$$\mathbb{R}^{f_s} \ni \mathbf{z}_s \mapsto \mathrm{Soft\!-\!Max} \circ \mathbf{W}\mathbf{z}_s \tag{1}$$

We call the W $\in \mathbb{R}^{\mathcal{C} \times f_s}$ as the "explanation matrix" – which can be obtained by minimizing some valid distance function (like Hellinger's distance) between distributions obtained as above and the probability distribution over classes that the complex model has been trained to map any input. Thus, we instantiate this novel mechanism for XAI, namely LIPEx.

Note how the matrix W in Equation 1 simultaneously gives for every feature a numerical measure of its importance for each possible class. We posit that it is important that in any explanation, it should be evident that most features deemed to be important for the predicted class are not so for the other classes - an idea that was recently formalized in Gupta & Arora (2020); Gupta et al. (2022) for the specific case of saliency maps. In our method, this property turns out to be emergent as a consequence of the more principled definition of explanation that we start from.

Figure 1 shows an example of our matrix explanation obtained for a text document. We observe how the explanation matrix is obtained for a specific document over a set of feature words. Note that for the first row (the top predicted class), the top 5 feature words detected for this instance ([feel, valued, joy, treasures, incredibly]) are *distinctly different* from the top features detected for the class in the second row, the one with the second highest probability predicted by the classifier. More examples like this can be found in Appendix D.5 and Appendix D.6 particularly focuses on examples where the predicted class and the true class are different. It is observed that there always arises a natural discrimination between features important for the different classes.

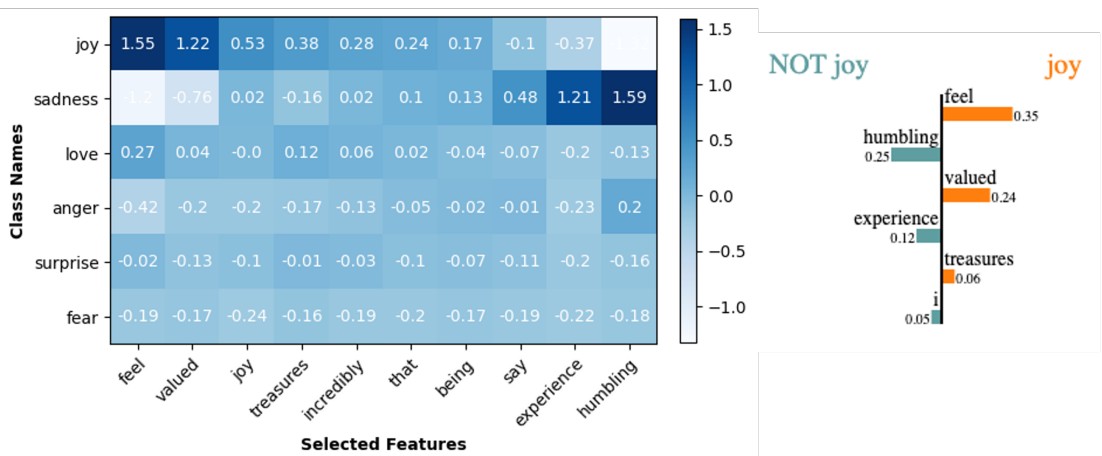

**Text**: *I have the joy of allowing kids to feel like the valued treasures that they are and to just have a blast being a kid alongside with them but can i just say its an incredibly humbling experience to have influence into a childs life and to know that what you do and say is being internalized.*

Figure 1: Example of comparison of explanation matrix obtained by LIPEx and the bar chart obtained by LIME on a text data from the Emotion dataset. For the LIPEx matrix, the class names on the left side are arranged in descending order of the predicted class probabilities. Examples of LIPEx explanation on image data is provided in Appendix D.3.

In the following, we summarize our contributions towards formalizing this idea of matrix-based explanations that match the distribution over classes predicted by a complex model.

**Novel Explanation Framework** In Section 3, we formally state our explanation approach, which can extract the relative importance of a set of features for every class under consideration beyond the true class. In the following tests, we demonstrate how such a multi-class explanation framework can be more useful for model understanding compared to existing state-of-the-art XAI methods.

**(Test 1) Evidence of LIPEx Replicating the Complex Model's Predicted Distribution Over Classes**   In Figure 2, we calculate the Total Variation (TV) distance of the output distribution of the obtained LIPEx explainer and the distribution output by the complex model for the same data and we show that over hundreds of randomly chosen test instances, the distance is overwhelmingly near 0. We show that this very necessary property holds over multiple models over text as well as images.

**(Test 2) Sanity Check of LIPEx's Sensitivity to Model Distortion**   In Figure 3 we distort a well-trained complex model by adding mean zero noise to the parameters in the last layer and measure how upon increasing the noise variance, the output probability distribution moves away in TV distance from the original prediction. We show that when LIPEx is implemented on the distorted models, our explainer's predicted distribution moves away from its original value almost identically. This sanity check is inspired by the arguments in Adebayo et al. (2018).

The above two tests give robust evidence that, indeed LIPEx is an accurate local approximator of the complex model while being dramatically simpler than the black-box predictor. To the best of our knowledge, such a strong model replication property is not known to be true for even the saliency methods, which can in-principle be called on different classes separately to get the relative importance among the different pixels for each class - albeit separately.

**(Test 3) Evidence of Changes in the Complex Model's Prediction Under LIPEx Guided Data Distortion**   In Table 1 and 2, we devise an ablation study guided by the "faithfulness" criteria as outlined in Atanasova et al. (2020). We establish that the top features detected by LIPEx are more important for the complex model than those detected by other XAI methods. We show this by demonstrating that when the top features are removed from the data and inference is done on this damaged data, then the new predicted class differs more from the original prediction when the removal is guided by what LIPEx deemed to be important than other XAI methods.

**(Test 4) Evidence of LIPEx Replicating the Complex Model's Class Prediction Under LIPEx Guided Data Distortion**   In Table 3, we demonstrate that for an overwhelming majority of data, upon removing their features deemed important by LIPEx, the new class predicted by the complex predictor is reproduced by the LIPEx model when presented with the same distorted data.

**(Test 5) Stability of LIPEx to Choosing Less and Only Near-Truth Perturbations**   In Figure 4, we demonstrate experiments that the features picked out by the LIPEx matrix are largely stable when the matrix is derived using only a few perturbation instances. We also show that this property is not true for LIME in the models we consider. Thus LIPEx is demonstrably more data efficient.

To put the above in context we recall that estimates were given in Agarwal et al. (2021) for how many perturbations around the true data are sufficient for LIME to produce reliable results - and this experiment of ours can be seen to corroborate that. Also, we recall that in works like Slack et al. (2020) it was pointed out that LIME's reliance on perturbations far from the true data creates a vulnerability that can be exploited to create adversarial attacks.

Note that we have restricted our attention to "intrinsic evaluations" of explanations, i.e., we only use calls to the model as a black-box for deciding whether the explanations obtained are meaningful as opposed to looking for external human evaluation. Both text and image data were used to evaluate our proposed approach. For text-based experiments we used 20Newsgroup [1] and Emotion [2] datasets. For image-based experiments, we have used the Imagenette[3] dataset with segments detected by "segment anything" [4].

Among the above experiments, LIPEx was compared against a wide range of state-of-the-art explanation methods for both text and image data, i.e., LIME (Ribeiro et al., 2016), Guided Backpropagation (Springenberg et al., 2014), Vanilla Gradients (Erhan et al., 2009), Integrated Gradients (Sundararajan et al., 2017), Deeplift (Shrikumar et al., 2016), Occlusion (Zeiler & Fergus, 2014), XRAI (Kapishnikov et al., 2019), GradCAM (Selvaraju et al., 2017), GuidedIG (Kapishnikov et al., 2021), BlurIG (Xu et al., 2020) and SmoothGrad (Smilkov et al., 2017a).

---

[1] http://qwone.com/~jason/20Newsgroups/
[2] https://huggingface.co/datasets/dair-ai/emotion
[3] https://github.com/fastai/imagenette
[4] https://segment-anything.com/

**Organization** In Section 2 we briefly overview related works in XAI. In Section 3 we give the precise loss function formalism for obtaining our explanation matrix, and in Section 4 all the tests will be given - comparing the relative benefits to other XAI methods. We conclude in Section 5. Appendices contain various details such as the precise pseudocode used in Section 4 (in Appendix C), the hyperparameter settings (in Appendix B), and further experimental data is given in Appendix D.

## 2 RELATED WORK

The work in Letham et al. (2015) is one of the first works that attempted to develop a classifier using rules and Bayesian analysis. In Ribeiro et al. (2016) a first attempt was made to describe explainability formally. The explanation can be made through an external explainer module, or a model can also be attempted to be made inherently explainable (Chattopadhyay et al., 2023). Post-hoc explainer strategy, as is the focus here, can be of different types, like (a) Ribeiro et al. (2016); Lundberg & Lee (2017) estimate feature importance for predicting a particular output, (b) counterfactual explanations (Wachter et al. (2017); Ustun et al. (2019); Rawal & Lakkaraju (2020)) determine if a feature $x$ was present in the input, then would the model have predicted output $y$, (c) contrastive approaches (Jacovi et al., 2021) describe why an ML model has predicted a particular output instead of another, or (d) Weinberger et al. (2023) and Crabbé & van der Schaar (2022) have recently proposed new XAI methods tuned to the case of unsupervised learning. In this work, we specifically focus on feature importance-based explanation techniques.

**Feature Importance-based Explanations** The study in Ribeiro et al. (2016) initiated the LIME framework which we reviewed in Section 1 as our primary point of motivation. Similarly, the work in Lundberg & Lee (2017) used a statistical sampling approach ("SHAP") to explain a classifier model in terms of human interpretable features. Lakkaraju et al. (2016) proposed a decision set-based approach to train a classifier that can be interpretable and accurate simultaneously - where a set of independent if-then rules defines a decision set. Ribeiro et al. (2018) proposed an anchor-based approach for explanation - where anchors were defined as a set of sufficient conditions for a particular local prediction.

Evaluation is a critical component in any explanation framework. The study in Doshi-Velez & Kim (2017) described important characteristics for the evaluation of explanation approaches. Evaluation criteria for explanations can broadly be categorized into two types, (a) criteria which measure how well the explainer module is able to mimic the original classifier and (b) criteria which measure the trustworthiness of the features provided by the explainer module, like the work in Qi et al. (2019) demonstrated the change in the prediction probability of a classifier with the removal of top $K$ features predicted by a saliency map explainer.

We note that in this work our tests done in Section 4 encompass both the above kinds of criteria.

Lastly, we note that in Sokol & Flach (2020) a tree based explanation was attempted which could directly work in the multiclass setting but to be able to compete LIME their method's computation cost can need to scale with the number of segments in an image. Also, in sharp contrast to our LIPEx proposal, it does not have the critical ability to explain/reproduce the predicted distribution of the given complex model.

## 3 OUR SETUP

Let $\mathcal{C} \in \{1, 2, 3, \ldots\}$ be the number of classes in the classification setup. Given any two probability vectors $\mathbf{p}, \mathbf{q} \in [0, 1]^{\mathcal{C}}$, $\sum_{i=1}^{\mathcal{C}} p_i = 1 = \sum_{i=1}^{\mathcal{C}} q_i$, we succinctly represent $\mathbf{p}, \mathbf{q}$ as being members of the simplex in $\mathcal{C}$−dimensions as $\mathbf{p}, \mathbf{q} \in \Delta^{\mathcal{C}}$.

**Classifier Setup** We aim to explain a classifier which can be described as a neural network $\mathcal{N}_{\mathbf{w}}$ (parameterized by weight $\mathbf{w}$) composed with a layer of soft-max so that the output of the composition is a probability distribution over the $\mathcal{C}$−classes. Thus we define the composed mapping,

$$\mathbf{f}_{\mathbf{w}} : \mathbb{R}^d \to \Delta^{\mathcal{C}}, \mathbf{x} \mapsto \text{Soft−Max} \circ \mathcal{N}_{\mathbf{w}}(\mathbf{x}) \tag{2}$$

This composed function $\mathbf{f}_{\mathbf{w}}$ in Equation 2 commonly would have been trained via the cross-entropy loss on a $\mathcal{C}$ class labeled data - and we assume only black-box access to it.

**The Feature Space for Explanations**  For a specific data $s$ (e.g., a piece of text), we denote the number of unique features (e.g., words) as $|s|$ and assume that there is a selected 'feature space' with $f_s$ features. Suppose special subsets of them, say $\mathcal{S}(s)$ and $\mathcal{S}_f(s)$ have been chosen and there is a map, say Select which does the feature selection for each of its domain points as per say Algorithm A.

$$\mathcal{S}(s) \subset \mathbb{R}^{|s|}, \ \mathcal{S}_f(s) \subset \mathbb{R}^{f_s} \ \& \ \text{Select} : \mathcal{S}(s) \to \mathcal{S}_f(s) \tag{3}$$

**The Local Explanation Matrix**  We explain $f_\mathbf{w}$'s behaviour around $s$ by a 'pseudo-linear model', $g_{s,\mathrm{W}}$ which is defined as,

$$\mathbf{g}_{s,\mathrm{W}} : \mathcal{S}_f(s) \to \Delta^\mathcal{C}, \ \mathbf{z}' \mapsto \text{Soft-Max} \circ \mathbf{W}\mathbf{z}' \tag{4}$$

with $\mathbf{W} \in \mathbb{R}^{\mathcal{C} \times f_s}$ being the "explanation matrix".

In the LIME setup (as well as in LIPEx), $\mathcal{S}(s) \subseteq \{0,1\}^{|s|}$ i.e. Boolean vectors are used to represent random ways of dropping one or more of the (unique) words for text data and pixels for image data. Hence, in such setups, the original input instance is represented as an all-ones vector, $\mathbf{1}_s \in \mathbb{R}^{|s|}$.

We assume that there is a pre-chosen function (say $T_s$) that maps "perturbations" of the data contained in the set $\mathcal{S}(s)$ to some $d$-dimensional embedding (like the BERT embeddings) which can be input to the original prediction model (Equation 2).

$$T_s : \mathcal{S}(s) \to \mathbb{R}^d \tag{5}$$

Note that, LIME seeks explanations using a linear function which would map the $\mathbf{z}'$ (as in Equation 4) to a real number which is a priori unbounded in sharp contrast to the explainer $\mathbf{g}_{s,\mathrm{W}}$ defined in Equation 4. Also note that the input dimensions $d$ of $f_\mathbf{w}$ and $f_s$ for $\mathbf{g}_{s,\mathrm{W}}$ could be very different and dependent on $s$ and typically, $f_s \lll d$. Eg., in standard LIME implementations for a classifier one often chooses $f_s = 6$ important features of the text $s$.

The space of all probability distributions admits various natural metrics and Hellinger distance has previously been used for feature selection in classification (Fu et al., 2020). Hellinger distance between two discrete distributions $\mathbf{p}, \mathbf{q}$ (on a set of $\mathcal{C}$ possible classes) is given as,

$$\mathrm{H}(\mathbf{p}, \mathbf{q}) := \frac{1}{\sqrt{2}} \cdot \sqrt{\sum_{c \in \mathcal{C}} \left(\sqrt{\mathbf{p}(c)} - \sqrt{\mathbf{q}(c)}\right)^2}$$

Apart from being an intuitive symmetric measure, squared Hellinger distance also offers other attractive features of being sub-additive, smaller than half of the KL divergence and always being within a quadratic factor of the Total Variation (TV) distance. (Canonne, 2020)[5]

Let $\tilde{\mathcal{S}}(s) \subset \mathcal{S}(s) (\subseteq \{0,1\}^{|s|})$ be a randomly sampled set of perturbations to be used for training. Passing it through the Select map (Equation 3) we obtain $\tilde{\mathcal{S}}_f(s) \subset \mathcal{S}_f(s) (\subset \mathbb{R}^{f_s})$ which are the feature representations of the perturbations. We posit that the outputs of the Select map would determine what the explainer $\mathbf{g}_{s,\mathrm{W}}$ in Equation 4 acts on. Further noting that the output of the embedding map $T_s$ in Equation 5 determines what the true predictor $f_\mathbf{w}$ gets as input, we consider the following empirical risk function corresponding to a distance function $\pi$ in $\mathbb{R}^{|s|}$,

$$\hat{\mathcal{L}}_{\mathrm{H}}(\mathbf{g}_{s,\mathrm{W}}, \tilde{\mathcal{S}}(s)) = \frac{1}{|\tilde{\mathcal{S}}(s)|} \sum_{\mathbf{x} \in \tilde{\mathcal{S}}(s)} \pi(\mathbf{1}_s, \mathbf{x}) \cdot \mathrm{H}^2\left(\mathbf{g}_{s,\mathrm{W}} \circ \text{Select}(\mathbf{x}), f_\mathbf{w} \circ T_s(\mathbf{x})\right) + \frac{\lambda}{2} \cdot \|\mathbf{W}\|_F^2 \tag{6}$$

where $\mathbf{1}_s$, the all-ones vector in $\mathbb{R}^{|s|}$. We choose $\pi(\mathbf{1}_s, \mathbf{x}) = 1 - \frac{\mathbf{x}^\top \mathbf{1}_s}{\|\mathbf{x}\| \cdot \|\mathbf{1}_s\|}$ for all our experiments. It is immediately interpretable that Equation 6 takes a $\pi$-weighted empirical average of the Hellinger distance squared between the true distribution over classes predicted by the complex classifier $f_\mathbf{w}$ and the distribution predicted by the explainer $\mathbf{g}_{s,\mathrm{W}}$ while the $\lambda$-term penalizes for using high weight explainers and hence promotes simplicity of $\mathbf{g}_{s,\mathrm{W}}$.

---

[5] Our experiments were tried with TV and they underperformed compared to the squared Hellinger metric.

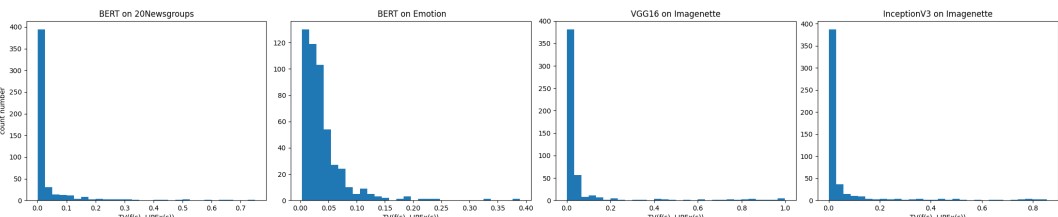

Figure 2: Histogram Statistics of TV Distance on Probability Distribution of Classes Between Classifier and LIPEx.

**Intuition for Good LIPEx Minima for Text Classifiers Being a ReLU Net** For intuition, consider explaining classification predictions by a ReLU net on a text data $s$ with $|s|$ unique words. Further suppose the classifier has been trained to accept $d(\geq |s|)$ word length texts in their TF-IDF representation. Thus the $T_s$ map (Equation 5) that lifts the perturbations of $s$ to the input space of the complex classifier can be imagined as a tall matrix of dimensions $d \times |s|$ whose top $|s| \times |s|$ block is a diagonal matrix giving the TF-IDF values for the words in this text and the rest of the matrix being zeros. Also, we note that the Select function can be imagined as a linear projection of the Boolean-represented perturbations into the subset of important features.

Further, any ReLU neural net is a continuous piecewise linear function Arora et al. (2018). Hence, except at the measure zero set of non-differentiable points, the function $\mathcal{N}_{\mathbf{w}}$ (Equation 2) is locally a linear function. Thus, for almost every input $\mathbf{z} \in \mathbb{R}^d$ there exists a (possibly small) neighbourhood of it where $\mathcal{N}_{\mathbf{w}} = \mathrm{W}_{\mathrm{net}}$ for some matrix $\mathrm{W}_{\mathrm{net}} \in \mathbb{R}^{\mathcal{C} \times d}$. It would be natural to expect that most true texts are not at the non-differentiable points of the net's domain and that $T_s$ maps small perturbations of the data into a small neighbourhood. Hence, for many perturbations $\mathbf{x} \in \mathbb{R}^{|s|}$, $\mathbf{f}_{\mathbf{w}}(T_s(\mathbf{x}))$, as it occurs in the loss in Equation 6, is a Soft-Max of a linear transformation (composition of the net and the $T_s$ map) of $\mathbf{x}$. Recall that this is exactly the functional form of the explainer $\mathbf{g}_{s,W}$ (Equation 4) given that the Select function can be represented as a linear map! Thus, we see that there is a very definitive motivation for this loss function to yield good locally linear explanations for ReLU nets classifying text.

## 4 RESULTS

At the very outset, we note the following salient points about our setup. *Firstly,* that for any data $s$ (say a piece of text or an image), when implementing LIPEx on it, we generate a set of 1000 perturbations of input instances. Then we chose features by taking a union set over the top-3 features of each possible class, which was returned by the "forward feature selection" method (reproduced in Algorithm A) called on the above perturbation data set. We recall that this feature selection algorithm is standard in LIME implementations[6]. Suppose this union has $f_s$ features - then for all computations to follow for $s$ we always stick with these $f_s$ features for LIPEx (and also always call LIME on $f_s$ number of features in comparison experiments). *Secondly,* we note that for the matrix returned by LIPEx (i.e. W in Equation 4) we shall define its "top$-k$" features as the features/columns of the matrix which give the $k-$highest entries by absolute value for the predicted class of that data.

**Reproducing the Distribution over Classes of the Complex Classifier** A key motivation for introducing the LIPEx framework was the need for the explanation framework to produce class distributions closely resembling those of the original classifier. Therefore, our initial emphasis is on investigating how much in Total Variation (TV) distance, the distribution over classes predicted by the obtained explainer is away from the one predicted for the same data by the complex model needing explanations. In Figure 2, we show the statistics of this TV distance for expeiments on both text (i.e. BERT on 20Newsgroups and BERT on Emotion) and image data (i.e. VGG16 and InceptionV3 on Imagenette). Figure 2 clearly shows that the distribution is highly skewed towards 0 over five hundred randomly sampled data over multiple modalities and state-of-the-art models. Note that the LIPEx loss (Equation 6) never directly optimized for the TV gap to be small and hence we posit that this is a strong test of performance that LIPEx passes.

---

[6]https://github.com/marcotcr/lime

**LIPEx Tracks Distortions of The Complex Model's Output Distribution**  This sanity check experiment is inspired by the studies in Adebayo et al. (2018). Here, we add mean-zero Gaussian noise to the trained complex model's last-layer weights and keep dialling up the noise variance till the model's accuracy is heavily damaged. At each noise level we compute the average over randomly sampled data, of the Total Variation distance between the output distribution of the damaged model and its original value and the same for the LIPEx's distribution for that model at respective inputs. We do text experiments with BERT on the Emotion dataset and image experiments with VGG16

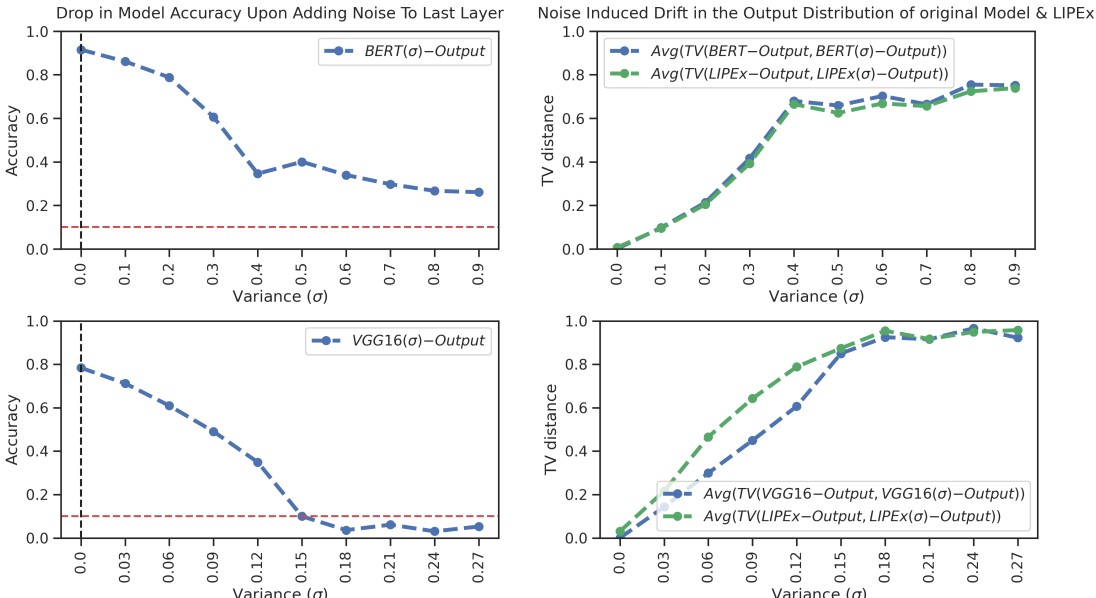

Figure 3: In the left image, we see how the model accuracy drops upon adding noise to the last layer weights and bias, a mean 0 Gaussian noise at different variances. This demonstrates that the maximum added noise is sufficient to distort the model highly. On right, we see how the data averaged TV distance between the output distribution of LIPEx for the data at the original complex model and the noise-distorted complex model tracks the same change in the complex model's output.

on the Imagenette dataset. In Figure 3, for any specific data, LIPEx−Output is the LIPEx's output distribution for the original model, LIPEx$(\sigma)$−Output is the LIPEx's output distribution for the distorted model at noise variance $\sigma$. BERT−Output, BERT$(\sigma)$−Output, VGG16−Output and VGG16$(\sigma)$−Output are defined similarly.

The right column of plots in Figure 3 demonstrates that as the model distorts, LIPEx's output moves away from the original in a remarkably identical fashion as the distorted model's output changes with respect to its original value.

**Importance of Top-K Features Detected by LIPEx**  A test of the correctness of determining any set of features to be important by an explanation method is that upon their removal from the original data and on presenting this modified/damaged input to the complex classification model it should produce a new predicted class than originally. We implement this test with text data in Table 1 and with image data in Table 2. We demonstrate that when the top features detected by LIPEx are removed from the data, the original model's predicted class changes substantially more than when the same is measured for many other XAI methods for the predicted class - and the amount of change is proportional to the number of top features removed. [7]

**LIPEx Reproduces the Complex Model's Class Predictions Under LIPEx Guided Data Damage** We posit that for a multi-class explainer as LIPEx, it is a very desirable sanity check that it should reproduce the underlying model's (new) predicted classes on the input when its top features are

---

[7]We use the code in Atanasova et al. (2020) to implement the gradient-based methods in Table 1, and the package https://github.com/PAIR-code/saliency to implement the saliency methods in Table 2

| Model & Dataset | Top-K | LIPEx | LIME | GuidedBack | Saliency | InputXGrad | Deeplift | Occlusion |
|---|---|---|---|---|---|---|---|---|
| BERT 20NewsGroups | K=1 | **0.781** | 0.777 | 0.387 | 0.387 | 0.38 | 0.38 | 0.45 |
| | K=2 | **0.857** | 0.841 | 0.477 | 0.477 | 0.48 | 0.48 | 0.543 |
| | K=3 | **0.897** | 0.856 | 0.517 | 0.517 | 0.52 | 0.52 | 0.59 |
| | K=4 | **0.909** | 0.881 | 0.517 | 0.517 | 0.523 | 0.523 | 0.627 |
| | K=5 | 0.908 | **0.912** | 0.553 | 0.553 | 0.57 | 0.57 | 0.653 |
| BERT Emotion | K=1 | **0.657** | 0.653 | 0.597 | 0.597 | 0.6 | 0.6 | 0.65 |
| | K=2 | **0.74** | 0.697 | 0.61 | 0.61 | 0.62 | 0.63 | 0.66 |
| | K=3 | **0.73** | 0.647 | 0.637 | 0.637 | 0.637 | 0.653 | 0.697 |
| | K=4 | **0.73** | 0.64 | 0.623 | 0.623 | 0.633 | 0.643 | 0.697 |
| | K=5 | **0.793** | 0.65 | 0.63 | 0.63 | 0.637 | 0.64 | 0.693 |

Table 1: Here, features refer to words. Upon removing top-K words detected by each of the XAI methods and doing re-prediction, we report the fraction of data on which the predicted class changes. We see that the words removed by LIPEx guidance more significantly impact the model's prediction than when guided by the other XAI methods. The complete experimental data with standard deviations can be seen in Table 6 in the appendix.

| Model & Dataset | Top-K | LIPEx | LIME | XRAI | GradCAM | GuidedIG | BlurIG | VanillaGrad | SmoothGrad | IG |
|---|---|---|---|---|---|---|---|---|---|---|
| VGG16 Imagenetee | K=2 | **0.763** | 0.74 | 0.713 | 0.69 | 0.717 | 0.713 | 0.68 | 0.747 | 0.703 |
| | K=3 | **0.82** | 0.78 | 0.77 | 0.763 | 0.75 | 0.787 | 0.753 | 0.817 | 0.747 |
| | K=4 | **0.867** | 0.793 | 0.793 | 0.79 | 0.793 | 0.807 | 0.807 | 0.843 | 0.773 |
| InceptionV3 Imagenette | K=2 | 0.673 | 0.63 | **0.693** | 0.653 | 0.663 | 0.647 | 0.657 | 0.65 | 0.637 |
| | K=3 | **0.753** | 0.713 | 0.7 | 0.697 | 0.67 | 0.703 | 0.653 | 0.683 | 0.707 |
| | K=4 | **0.773** | 0.767 | 0.74 | 0.72 | 0.713 | 0.717 | 0.72 | 0.74 | 0.733 |

Table 2: Here, features refer to image segments which were gotten by Segment Anything. LIPEx and LIME can be used to directly get a weight for each segment while for the saliency-based methods a segment's importance is determined as the sum of the weights assigned to its pixels. In the table above we can see that the fraction of data on which label prediction changes under deletion of top features detected by LIPEx is consistently higher than for other XAI methods. The complete experimental data with the standard deviation can be found in the Table 7 in the appendix.

removed. In Table 3, we show with text as well as image data, that this class prediction matching holds for the LIPEx explainer for an overwhelming majority of data.

**Evidence for Data Efficiency of LIPEx as Compared to LIME** Since LIPEx and LIME, both are perturbation based methods, a natural question arises if LIPEx is more data-efficient, or in other words can its top features detected be stable if only a few perturbations close to the true data are allowed. In this test, we show that not only is this true, but also that (a) LIPEx's top features can at times even remain largely invariant to reducing the perturbations and also that (b) the difference with respect to LIME in the list of top features detected, is maintained when the allowed set of perturbations are increasingly constrained to be few and near the true data. Our comparison method is specified precisely as Algorithm C in the Appendix and we sketch it here as follows.

When in the setting with unrestricted perturbations, we infer two lists of top features, one from the row of the predicted class of the matrix (i.e. $W$) returned by LIPEx and another from LIME's weight vector for the same class — say $\mathrm{LIPEx{-}List{-}s}$ and $\mathrm{LIME{-}List{-}s}$ respectively. Next, we parameterize the restriction on the allowed perturbations by the maximum angle $\delta$ that any Boolean vector representing the perturbation is allowed to subtend with respect to the all-ones vector that represents the input data.

We use the default set of perturbations in a LIME implementation as a baseline [8] and at different $\delta$, we use only the $\delta{-}$restricted subset of the perturbations to compute (for the model predicted class) the top features returned by the LIPEx matrix and the LIME, say $\delta{-}\mathrm{LIPEx{-}List{-}s}$ and $\delta{-}\mathrm{LIME{-}List{-}s}$ respectively. For quantifying the dissimilarities between these lists of top features measured by the two methods, we compute the following Jaccard indices and average the results on 100 randomly chosen instances.

---

[8] In the LIME code, they choose perturbations of 5000 for text data and 1000 for image data.

| Model & Dataset | Modality | Top1 | Top2 | Top3 | Top4 | Top5 |
|---|---|---|---|---|---|---|
| BERT (20Newsgroups) | Text | 0.90 (±0.041) | 0.85(±0.024) | 0.79(±0.039) | 0.71(±0.033) | 0.70(±0.005) |
| BERT (Emotion) | Text | 0.89(±0.022) | 0.84(±0.025) | 0.84(±0.017) | 0.82(±0.037) | 0.74(±0.033) |
| VGG16(Imagenette) | Image | 0.80(±0.046) | 0.73(±0.034) | 0.73(±0.034) | 0.73(±0.025) | 0.70(±0.075) |
| InceptionV3 (Imagenette) | Image | 0.90(±0.051) | 0.78(±0.044) | 0.75(±0.015) | 0.74(±0.013) | 0.69(±0.035) |

Table 3: In this table, for each model and data combination, we give the fraction of data (over 100 random samples) over which the new class predicted by the complex model matches the new prediction by the LIPEx for the same model, upon removing from the data its top features as determined by LIPEx. We can observe that post this distortion on the data, the class labels from the complex model match those from the simple explainer for a significant majority of the instances.

$$J_{s,\delta,\text{LIME}} := \left| \frac{\delta-\text{LIME-List-s} \ \cap \ \text{LIME-List-s}}{\delta-\text{LIME-List-s} \ \cup \ \text{LIME-List-s}} \right|, \ J_{s,\delta,\text{LIPEx}} := \left| \frac{\delta-\text{LIPEx-List-s} \ \cap \ \text{LIPEx-List-s}}{\delta-\text{LIPEx-List-s} \ \cup \ \text{LIPEx-List-s}} \right|$$

$$J_{s,\delta-\text{LIPEx-vs-LIME}} := \left| \frac{\delta-\text{LIPEx-List-s} \ \cap \ \text{LIME-List-s}}{\delta-\text{LIPEx-List-s} \ \cup \ \text{LIME-List-s}} \right|$$

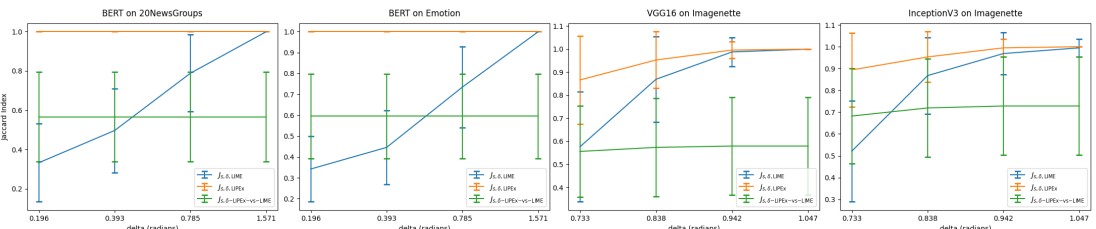

Figure 4: It can be observed that $J_{s,\delta,\text{LIPEx}}$ (the orange line) is very stable compared to that of LIME despite the allowed perturbations being made constrained. The difference in LIPEx's features w.r.t LIME is also maintained. The number of points considered at different $\delta$ is given in Table 5 in the appendix.

From Figure 4, we can infer that at all levels of constraint on the data at least $50\%$ of the top features detected by our LIPEx are different from LIME. *Secondly,* $J_{s,\delta,\text{LIME}}$ (averaged) rapidly falls as the number of training data allowed near the input instance is decreased. Thus its vividly revealed that the features detected by LIME are significantly influenced by those perturbations that are very far from the true text.

*Lastly,* and most interestingly, we note that the curve for $J_{s,\delta,\text{LIPEx}}$ (the top orange line) is very stable to using only a few perturbations which subtend a low angle with the true text. Hence the top features detected by our explanation matrix are not only important (as demonstrated in the previous two experiments) – but can also be computed very data efficiently.

## 5    CONCLUSION

In this work, we proposed a novel explainability framework, LIPEx, that when implemented in a classification setting, in a single training gives a weight assignment for all the possible classes for an input with respect to a chosen set of features. Unlike other XAI methods it is designed to locally approximate the probabilities assigned to the different classes by the complex model - and this was shown to bear out in experiments over text and images - and it withstood ablation tests. Our experiments, showed that the LIPEx proposal provides more trustworthy and data-efficient explanations compared to multiple other competing methods across various data modalities.

We note that our XAI loss, Equation 6, can be naturally generalized to other probability metrics like the KL divergence. Our studies strongly motivate novel future directions about not only exploring the relative performances between these options but also about obtaining guarantees on the quality of the minima of such novel loss functions.

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

## A  FEATURE SELECTION

---

Algorithm 1: Forward feature selection

---

**Require:** data X, target $Y$, number of features $k$

1:                                                 $\triangleright$ $X \in \mathbb{R}^{\#\text{Perturbations}\times\#\text{Unique}-\text{Words}}$

2:                                                 $\triangleright$ $Y \in \mathbb{R}^{\#\text{Perturbations}\times\#\text{Num}-\text{Classes}}$

**Ensure:** set of indices of the selected features `Sel_feats`

3:  `Sel_feats` $\leftarrow \{\}$

4: **for** $y$ in $Y$ **do**                                  $\triangleright$ $y \in \mathbb{R}^{\#\text{Perturbations}\times 1}$

5:     $f \leftarrow$ initialize selection model of Ridge regression

6:     `Current_sel_feats` $\leftarrow \{\}$

7:     `All_feats` $\leftarrow \{1, 2, .., \text{len}(X[0])\}$        $\triangleright$ `len(X[0])` =Number Of Unique Words in $X$

8:     **for** $i \leftarrow 1$ to $k$ **do**

9:         `best_idx` $\leftarrow 0$

10:        `best_score` $\leftarrow -\infty$

11:        **for** $j \in (\text{All\_feats} \smallsetminus \text{Sel\_feats})$ **do**

12:           $f \leftarrow$ `f.fit(X[:, Sel_feats` $\cup \{$`j`$\}$`], y)`

13:        $\triangleright$ `f.fit()` is used to train f, where the loss function is the linear least squares with l2-norm.

14:           `score` $\leftarrow$ evaluate $f$ with performance metric of $R^2$

15:           **if** `score` $>$ `best_score` **then**

16:              `best_idx` $\leftarrow$ `j`

17:              `best_score` $\leftarrow$ `score`

18:           **end if**

19:        **end for**

20:        `Current_sel_feats` $\leftarrow$ `Current_sel_feats` $\cup \{$`best_idx`$\}$

21:     **end for**

22:     `Sel_feats` $\leftarrow$ `Sel_feats` $\cup$ `Current_sel_feats`

23: **end for**

24: **return** `Sel_feats`

---

## B  LIPEX HYPERPARAMETER SETTINGS

Hyperparameter search was conducted over a small set of randomly selected data of each of the types mentioned below to decide on the following choices.

| Learning rate | $\lambda$ | Batch size |
|---|---|---|
| 0.01 | 0.001 | 128 |

Table 4: LIPEx Hyperparameter Settings

Note that $\lambda$ in above refers to the regularizer in the loss in equation 6.

## C  Pseudocode for the Quantitative Comparison Between LIPEx and LIME's Detected Important Features (as given in Section 4)

---

**Algorithm 2: LIME vs LIPEx w.r.t Angular Spread of the Perturbations About The True Data**

---

**Require:** $k$ = number of top features to be used for comparing LIME and LIPEx
**Require:** A set $\mathcal{S}$ of randomly sampled class labelled data at which the comparison is to be done
**Require:** $f^*$ = the trained predictor that needs explanations.
**Require:** $\delta-$List of all the angular deviations about the true data at which the LIPEx vs LIME comparison is to be done

1: **for** s $\in \mathcal{S}$ **do**
2:    Compute LIPEx$-$List$-$s = top-$k$ features of $s$ w.r.t its predicted class, as detected by the LIPEx matrix using the standard set of Boolean vectors/perturbations w.r.t the all-ones representation of $s$.
3:    Compute LIME$-$List$-$s = top-$k$ features of $s$ w.r.t its predicted class, as detected by LIME using the standard set of Boolean vectors/perturbations w.r.t the all-ones representation of $s$ - on the same set of features as used in the previous step.
4:                 ▷ **Note that the above two lists of "important" features do not depend on $\delta$,**
5:                 ▷ **We shall use both as reference lists for the different comparisons to follow.**
6:             ▷ **The list of features used above will be held fixed in the computations below.**
7:    **for** $\delta \in \delta - $List **do**
8:        Compute $\delta-$LIPEx$-$List$-$s = top-$k$ features of $s$ w.r.t its predicted class, as detected by the LIPEx matrix using only those Boolean vectors/perturbations which are within an angle of $\delta$ w.r.t the all-ones representation of $s$.
9:        Compute $\delta-$LIME$-$List$-$s = top-$k$ features of $s$ w.r.t its predicted class, as detected by LIME using only those Boolean vectors/perturbations which are within an angle of $\delta$ w.r.t the all-ones representation of $s$
10:
11:        Compute the Jaccard Index, $J_{s,\delta-\text{LIPEx-vs-LIME}} := \left| \frac{\delta-\text{LIPEx}-\text{List}-\text{s} \cap \text{LIME}-\text{List}-\text{s}}{\delta-\text{LIPEx}-\text{List}-\text{s} \cup \text{LIME}-\text{List}-\text{s}} \right|$
12:        Compute the Jaccard Index, $J_{s,\delta,\text{LIME}} := \left| \frac{\delta-\text{LIME}-\text{List}-\text{s} \cap \text{LIME}-\text{List}-\text{s}}{\delta-\text{LIME}-\text{List}-\text{s} \cup \text{LIME}-\text{List}-\text{s}} \right|$
13:        Compute the Jaccard Index, $J_{s,\delta,\text{LIPEx}} := \left| \frac{\delta-\text{LIPEx}-\text{List}-\text{s} \cap \text{LIPEx}-\text{List}-\text{s}}{\delta-\text{LIPEx}-\text{List}-\text{s} \cup \text{LIPEx}-\text{List}-\text{s}} \right|$
14:    **end for**
15: **end for**
16: Plot $\left( \frac{1}{|\mathcal{S}|} \cdot \sum_{s \in \mathcal{S}} J_{s,\delta-\text{LIPEx-vs-LIME}} \right)$ vs $\delta$
17: Plot $\left( \frac{1}{|\mathcal{S}|} \cdot \sum_{s \in \mathcal{S}} J_{s,\delta,\text{LIME}} \right)$ vs $\delta$
18: Plot $\left( \frac{1}{|\mathcal{S}|} \cdot \sum_{s \in \mathcal{S}} J_{s,\delta,\text{LIPEx}} \right)$ vs $\delta$

---

# D ADDITIONAL EXPERIMENTS

## D.1 $\delta$ EFFECT ON JACCARD EXPERIMENT

| For Text data | | | | |
|---|---|---|---|---|
| $\delta$ (radians) | $\frac{\pi}{16}$ | $\frac{\pi}{8}$ | $\frac{\pi}{4}$ | $\frac{\pi}{2}$ |
| number of perturbation points | 138 | 659 | 2383 | 5000 |
| **For Image data** | | | | |
| $\delta$ (radians) | $\frac{7\pi}{30}$ | $\frac{8\pi}{30}$ | $\frac{9\pi}{30}$ | $\frac{10\pi}{30}$ |
| number of perturbation points | 228 | 774 | 994 | 1000 |

Table 5: The effect of $\delta$ on the number of perturbation points, result averaged on 100 input instances.

Note that when $\delta$ decreases, while the amount of allowed perturbations falls, the similarity measure $\pi$ in equation 6 increases.

## D.2 ADDITIONAL DATA FOR THE DEMONSTRATION IN FIGURE 1

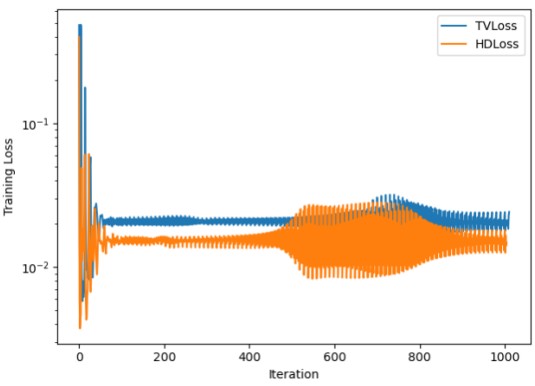

Figure 5: Comparison of the progress of training on the Hellinger distance based LIPEx loss, as given in equation 6, (its training curve being labelled as "HDLoss" above) and a natural Total Variation distance analogue of it (its training curve labelled as "TVLoss" above), for the text data in Figure 1

## D.3 LIPEx ON IMAGE

Each class/row of our explanation matrix would contain a weight corresponding to the importance of a common set of features/super-pixels for that class. The figure below shows the part of the matrix corresponding to the top 3 classes detected for this image i.e. "Burmese_mountain_dog", "Entlebucher" and "Appenzeller" and the top$-4$ features deemed to be important for the predicted class i.e. "Burmese_mountain_dog". Thus we see how LIPEx successfully "localized" the dog as being determinant to the predictions rather than the cat which is also prominent in this picture.

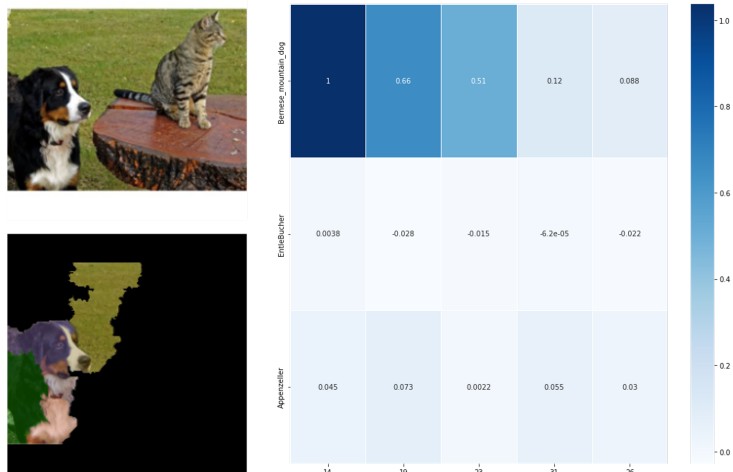

Figure 6: Example of (the most important part) of the matrix returned by the LIPEx method on an image. See Figure 7 how the top 5 segments detected for the image patch together. The corresponding LIME answer is visualized in Figure 8 - and we can see how it prioritized image segments unrelated to the dog.

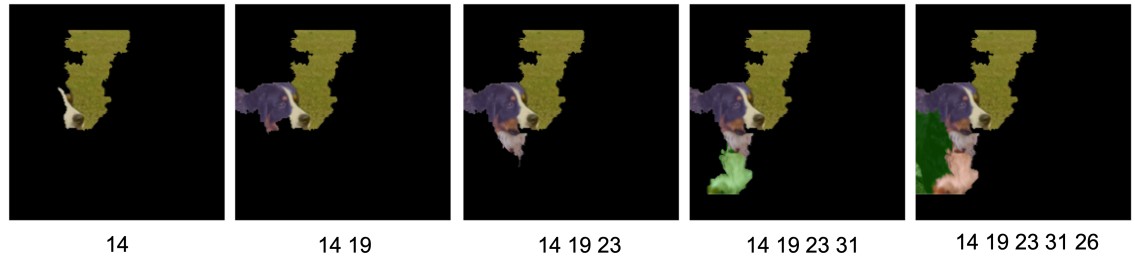

Figure 7: The top 5 image segments deemed to be important by LIPEx for Inception-V3 to classify the image in Figure 6 as a "Burmese_mountain_dog"

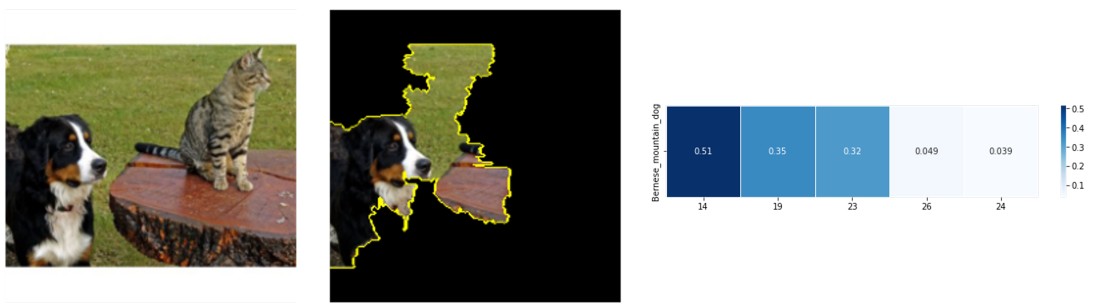

Figure 8: The weight vector over the features as returned by LIME

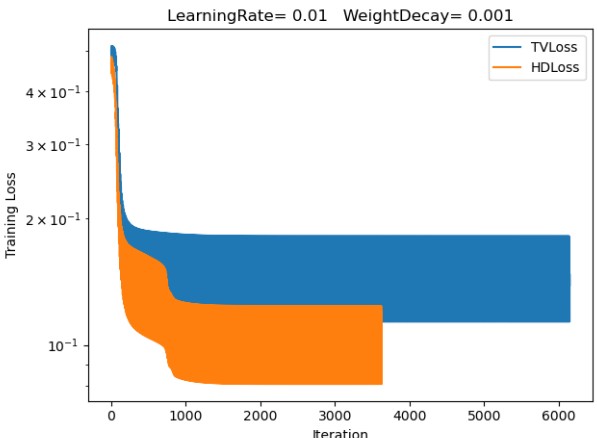

Figure 9: The LIPEx result for the image data in Figure 7 corresponds to the endpoint of minimizing the loss given in equation 6. (its training curve being labelled as "HDLoss" above). The training curve labelled as "TVLoss" corresponds to training on an analogous loss as in equation 6 but with the metric used in probability space being Total Variation.

## D.4    More Details About the Ablation Studies in Section 4

| Method | Top3 | Top4 | Top5 |
|---|---|---|---|
| **Vgg16** | | | |
| **LIPEx** | **0.763**(±0.026) | **0.82**(±0.014) | **0.867**(±0.017) |
| LIME | 0.74(±0.014) | 0.78(±0) | 0.793(±0.009) |
| XRAI | 0.713(±0.025) | 0.77(±0.008) | 0.793(±0.026) |
| GradCAM | 0.69(±0.016) | 0.763(±0.026) | 0.79(±0.043) |
| GuidedIG | 0.717(±0.017) | 0.75(±0.016) | 0.793(±0.009) |
| BlurIG | 0.713(±0.009) | 0.787(±0.017) | 0.807(±0.017) |
| Vanilla_Grad | 0.68(±0.022) | 0.753(±0.005) | 0.807(±0.012) |
| SmoothGrad | 0.747(±0.019) | 0.817(±0.005) | 0.843(±0.005) |
| Integrated_Grad | 0.703(±0.021) | 0.747(±0.029) | 0.773(±0.017) |
| **InceptionV3** | | | |
| **LIPEx** | 0.673(±0.005) | **0.753**(±0.005) | **0.773**(±0.017) |
| LIME | 0.63(±0.014) | 0.713(±0.046) | 0.767(±0.034) |
| XRAI | **0.693**(±0.017) | 0.7(±0.029) | 0.74(±0.062) |
| GradCAM | 0.653(±0.005) | 0.697(±0.017) | 0.72(±0.036) |
| GuidedIG | 0.663(±0.046) | 0.67(±0.051) | 0.713(±0.04) |
| BlurIG | 0.647(±0.041) | 0.703(±0.034) | 0.717(±0.029) |
| Vanilla_Grad | 0.657(±0.012) | 0.653(±0.026) | 0.72(±0.051) |
| SmoothGrad | 0.65(±0.024) | 0.683(±0.049) | 0.74(±0.037) |
| Integrated_Grad | 0.637(±0.019) | 0.707(±0.025) | 0.733(±0.04) |

Table 6: Ablation study of Vgg16 and InceptionV3, results averaged on three times run, each run over randomly chosen 100 images from the ImageNette validation dataset. Here, features refer to image segments (via Segment Anything) weighted by LIME/LIPEx or pixel importances determined by saliency methods.

| Method | Top1 | Top2 | Top3 | Top4 | Top5 |
|---|---|---|---|---|---|
| | **BERT on 20Newsgroups** | | | | |
| **LIPEx** | **0.781**($\pm$0.047) | **0.857**($\pm$0.036) | **0.897**($\pm$0.016) | **0.909**($\pm$0.026) | 0.908($\pm$0.032) |
| LIME | 0.777($\pm$0.027) | 0.841($\pm$0.031) | 0.856($\pm$0.045) | 0.881($\pm$0.011) | 0.912($\pm$0.021) |
| GuidedBack | 0.387($\pm$0.049) | 0.477($\pm$0.082) | 0.517($\pm$0.074) | 0.517($\pm$0.054) | 0.553($\pm$0.042) |
| Saliency | 0.387($\pm$0.049) | 0.477($\pm$0.082) | 0.517($\pm$0.074) | 0.517($\pm$0.054) | 0.553($\pm$0.042) |
| Input_G | 0.38($\pm$0.054) | 0.48($\pm$0.079) | 0.52($\pm$0.071) | 0.523($\pm$0.05) | 0.57($\pm$0.029) |
| Deeplift | 0.38($\pm$0.054) | 0.48($\pm$0.079) | 0.52($\pm$0.071) | 0.523($\pm$0.05) | 0.57($\pm$0.029) |
| Occlusion | 0.45($\pm$0.045) | 0.543($\pm$0.066) | 0.59($\pm$0.082) | 0.627($\pm$0.065) | 0.653($\pm$0.063) |
| | **BERT on Emotion** | | | | |
| **LIPEx** | **0.657**($\pm$0.021) | **0.74**($\pm$0.037) | **0.73**($\pm$0.028) | **0.73**($\pm$0.029) | **0.793**($\pm$0.024) |
| LIME | 0.653($\pm$0.017) | 0.697($\pm$0.041) | 0.647($\pm$0.037) | 0.64($\pm$0.045) | 0.65($\pm$0.008) |
| GuidedBack | 0.597($\pm$0.029) | 0.61($\pm$0.029) | 0.637($\pm$0.009) | 0.623($\pm$0.009) | 0.63($\pm$0.008) |
| Saliency | 0.597($\pm$0.029) | 0.61($\pm$0.029) | 0.637($\pm$0.009) | 0.623($\pm$0.009) | 0.63($\pm$0.008) |
| Input_G | 0.6($\pm$0.033) | 0.62($\pm$0.029) | 0.637($\pm$0.009) | 0.633($\pm$0.009) | 0.637($\pm$0.005) |
| Deeplift | 0.6($\pm$0.033) | 0.63($\pm$0.028) | 0.653($\pm$0.012) | 0.643($\pm$0.012) | 0.64($\pm$0.008) |
| Occlusion | 0.65($\pm$0.016) | 0.66($\pm$0.022) | 0.697($\pm$0.012) | 0.697($\pm$0.005) | 0.693($\pm$0.017) |

Table 7: Ablation study on Text dataset by removing TopK features detected by explainable methods and doing re-prediction, each experiment was independently repeated three times on randomly chosen 100 text instances.

### D.5 COMPARISON BETWEEN THE EXPLANATIONS FOUND BY LIPEX AND LIME

Figure 10, 11 vividly demonstrate the fine-grained explanation that is obtained by the LIPEx matrix as opposed to the LIME's explanation (right bar) on the same number of feature sets. The input instances in Figure 10 and 11 are randomly chosen from the Emotion dataset. The explanatory matrix generated by LIPEx makes it easy to see the relationship between the same feature and different categories.

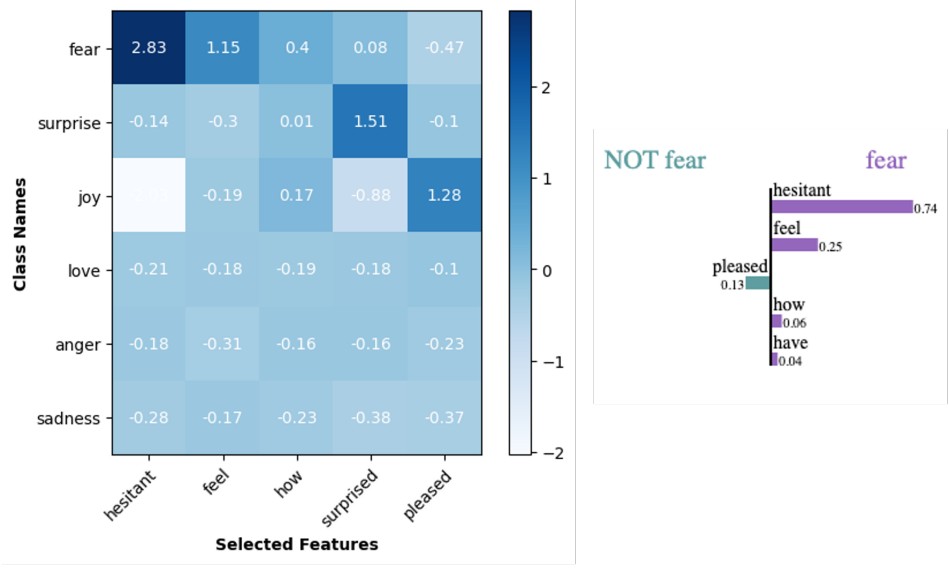

**Text:** *i feel like in the last year especially i ve gone from a girl to a woman and despite how hesitant i have always been about getting older next year i will be twenty four i am surprised at how pleased i am to have done so.*

Figure 10:

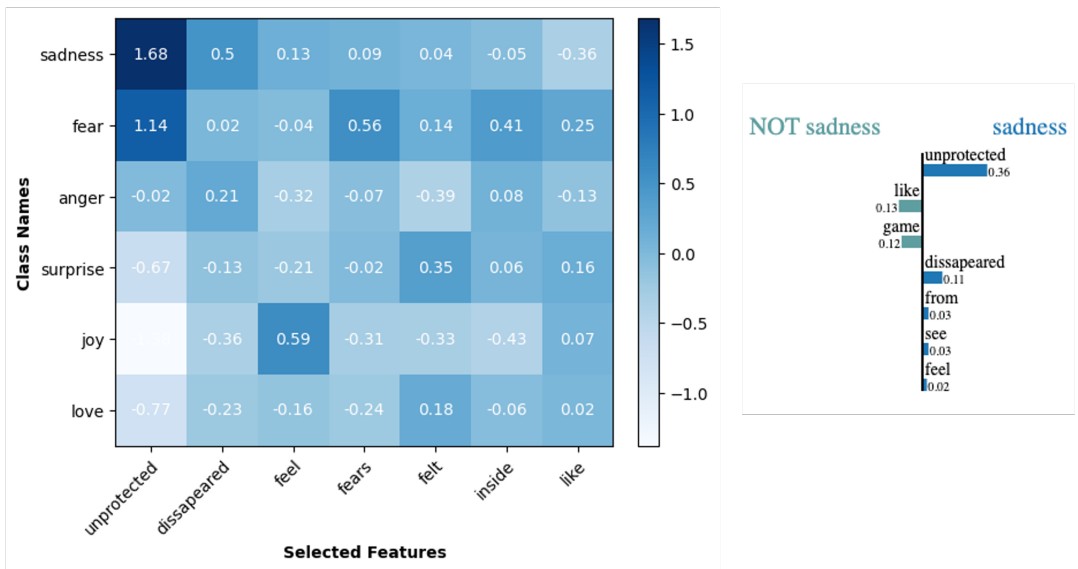

**Text:** *i feel inside this life is like a game sometimes then you came around me the walls just dissapeared nothing to surround me keep me from my fears im unprotected see how ive opened up youve made me trust coz ive never felt like this before im naked around you does it show.*

Figure 11:

### D.6 COMPARISON BETWEEN THE EXPLANATIONS FOUND BY LIPEx AND LIME FOR TEXT DATA WHERE PREDICTED CLASS AND THE TRUE CLASS ARE DIFFERENT

Figure 12, 13, 14, 15 demonstrate the fine-grained explanation that is obtained by the LIPEx matrix as opposed to the LIME's explanation on the same feature set for the predicted class - which is different than the true class for these instances. The input instances in Figure 12, 13, 14 and 15 are chosen from 20Newsgroups.

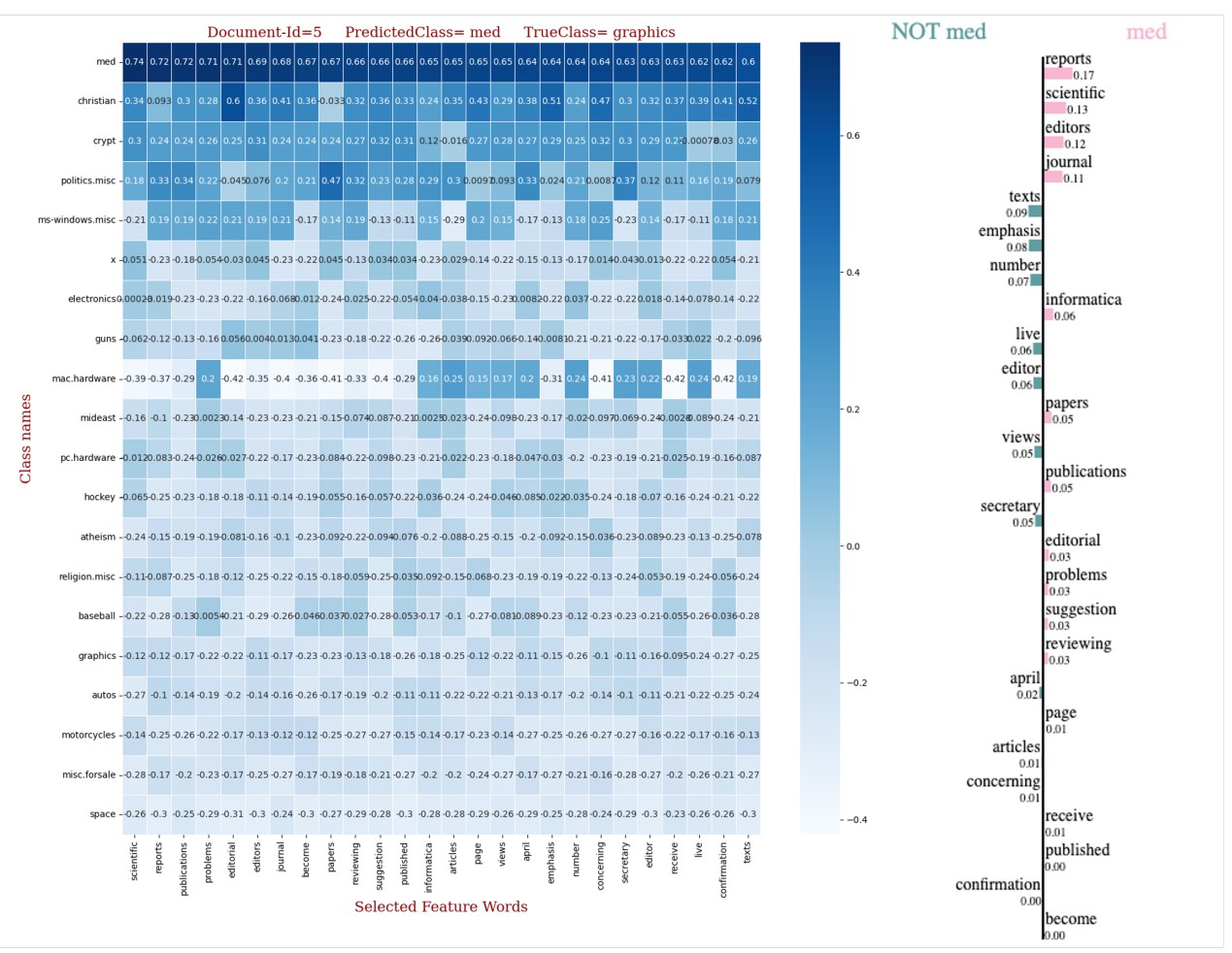

Figure 12:

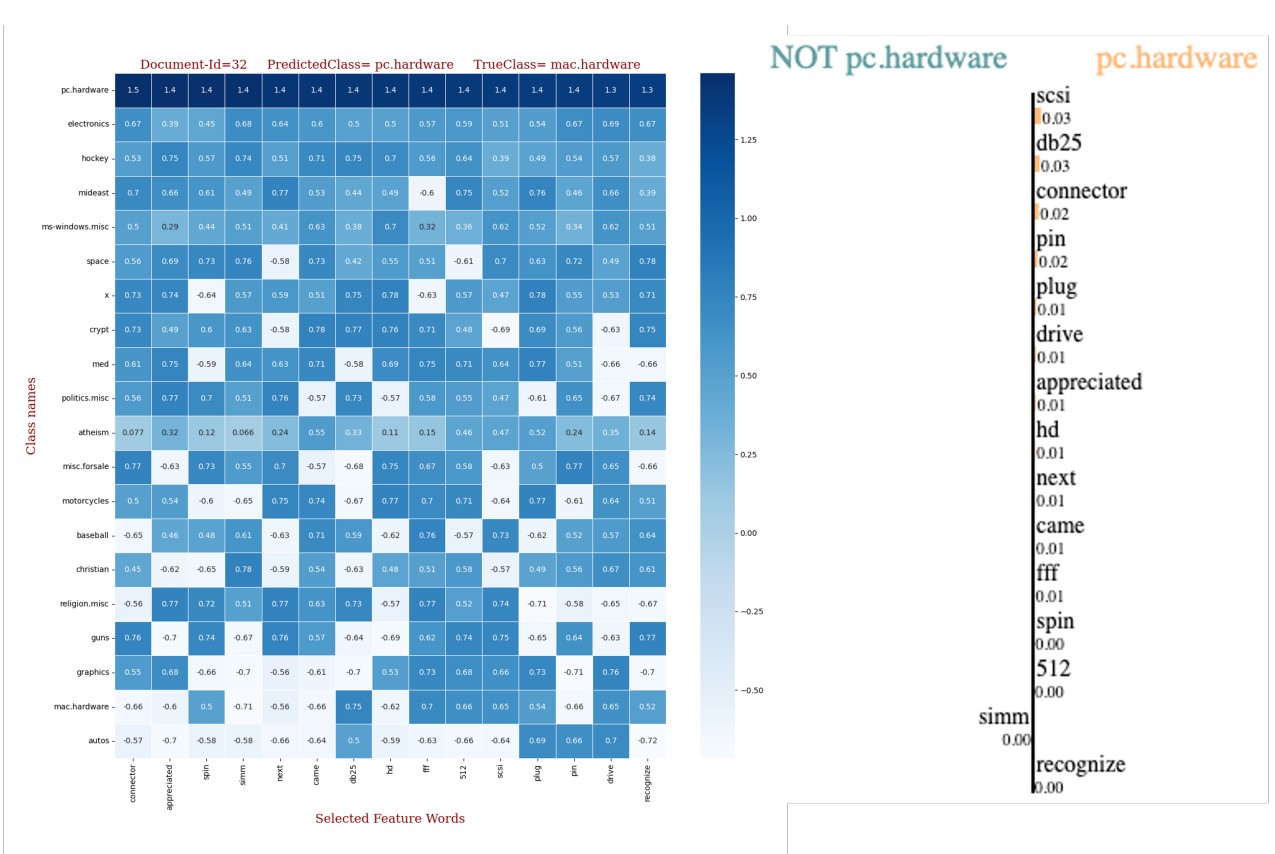

Figure 13:

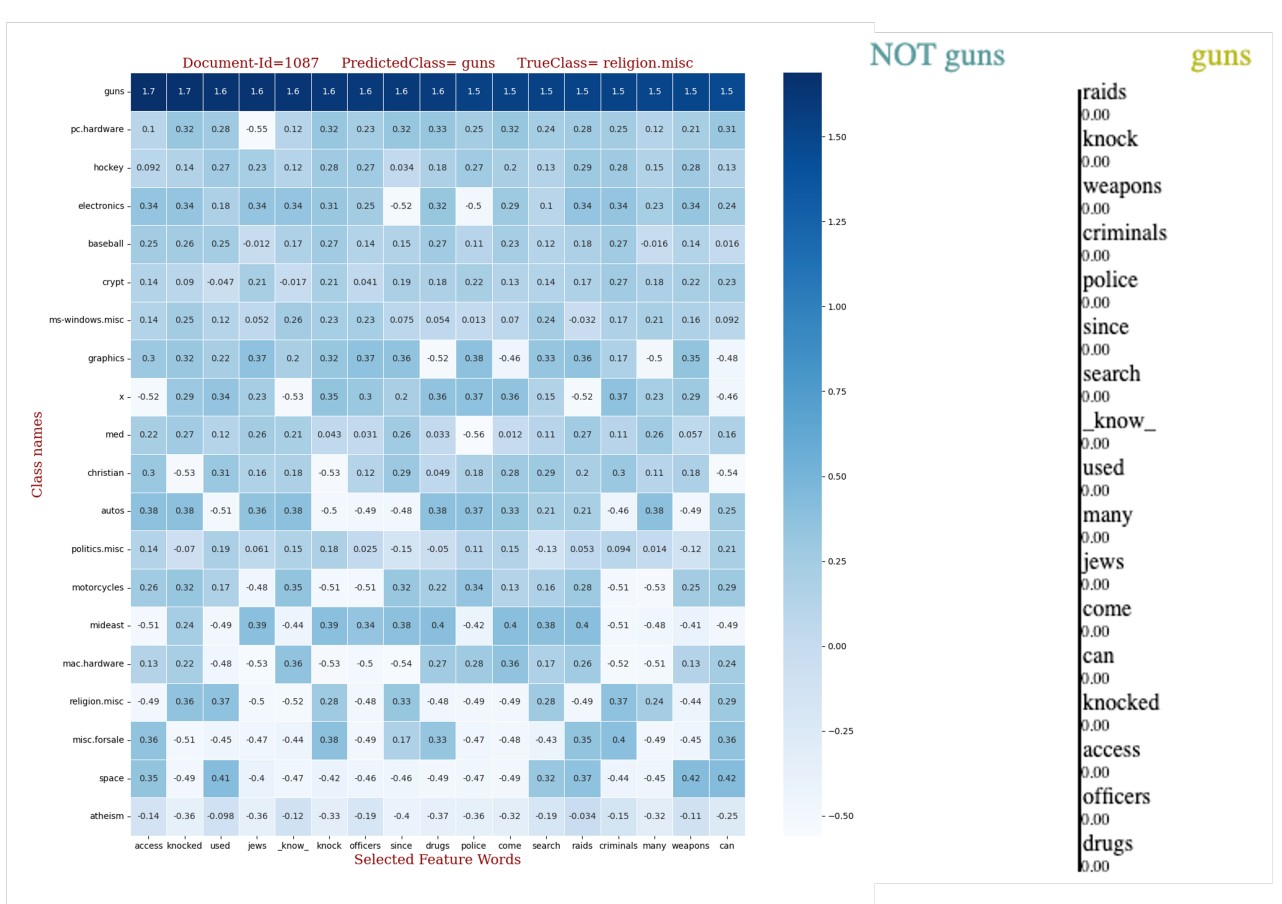

Figure 14:

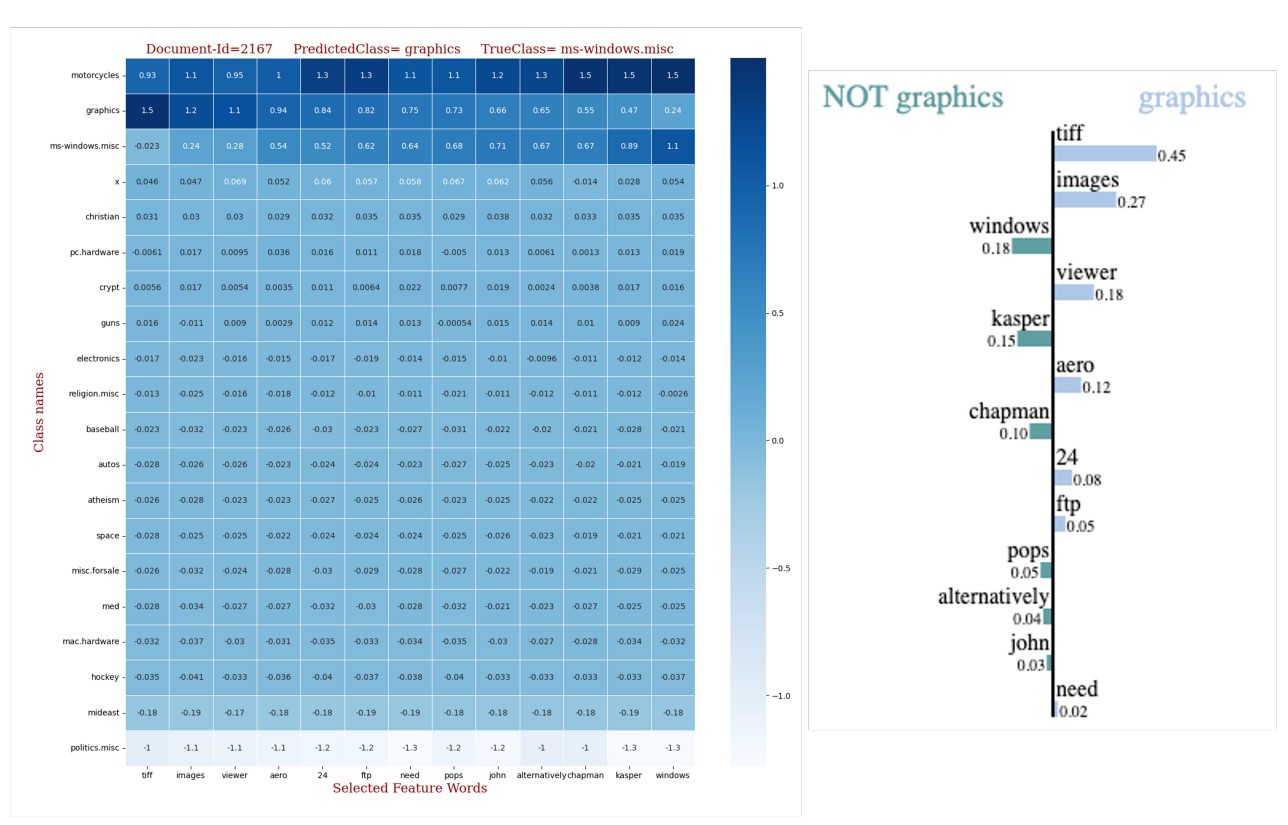

Figure 15:

