# OpenReview forum: "LIPEx -- Locally Interpretable Probabilistic Explanations -- To Look Beyond The True Class"
_ICLR.cc/2024/Conference — ICLR 2024 Conference Withdrawn Submission_

### Official Review · Reviewer_BvXE · 2023-10-23

**Soundness:** 2 fair
**Presentation:** 4 excellent
**Contribution:** 1 poor
**Rating:** 3
**Confidence:** 5

**Summary:**

Summary:

The paper proposes a new perturbation-based multi-class explanation framework, LIPEx. While LIME learns a regression model to identify the important features to the prediction of one class, LIPEx aims to learn an explanation matrix that highlights the features crucial to the predictions of all classes. Experimental results demonstrate that LIPEx provides explanations that are more faithful than those produced by LIME and some other popular XAI methods. Additionally, it has been shown that LIPEx operates with greater efficiency than LIME.

**Strengths:**

1. The paper is well-structured and well-written, making it easy to understand.

2. The research question, which revolves around determining the importance of different features for all possible class likelihoods estimated by a classifier, is indeed crucial and deserves further research.

**Weaknesses:**

1. Missing many related works.

In the statement of paper, "The full spectrum of feature influence on each class at a particular data point can help to understand how well the model has been trained to discriminate a particular class from the rest. However, existing explanation frameworks do not provide any clue on the aforementioned issue," the authors may have overstated the absence of existing frameworks that address this problem.

While it may be true that a comprehensive solution providing a full spectrum of feature influence on each class may not exist yet, several works have made attempts to explain not just the top-1 class, but also other classes. These include various counterfactual and contrastive explanation frameworks, which provide insights into "why A but not B" and "why B but not A" scenarios. For instance, consider the following papers:

- "Contrastive Explanations for Model Interpretability" (Jacovi et al., 2021)

- "Counterfactual Explanations without Opening the Black Box: Automated Decisions and the GDPR" (Wachter et al., 2018)

- "Counterfactual Visual Explanations" (Goyal et al., 2019)

- "SCOUT: Self-aware Discriminant Counterfactual Explanations" (Wang & Vascon, 2020)

- "Two-Stage Holistic and Contrastive Explanation of Image Classification" (Xie et al., 2023)

- "CoCoX: Generating Conceptual and Counterfactual Explanations via Fault-Lines" (Akula  et al., 2020)

- "Why Not Other Classes?: Towards Class-Contrastive Back-Propagation Explanations" (Wang et al., 2022)

These studies are not referenced or evaluated in the paper under review.

2.

In the paper's faithfulness evaluation using removal-based methods (which involves the elimination of top features identified by explanations), two concerns arise:

2a. The stronger baseline of perturbation-based explanation methods, such as RISE (as discussed in this [paper](https://arxiv.org/pdf/1806.07421.pdf)), is not taken into account. The current experiment only considers LIME and Occlusion, which limits the comprehensiveness of the findings.

2b. The current evaluation is solely focused on the top-1 prediction class. However, the authors claim that their paper's goal is to provide an explanation framework for all classes. As such, evaluations should also be extended to other classes — at least for the class with the second highest probability, as done in this [study](https://openreview.net/forum?id=X5eFS09r9hm). When evaluating the class with the second highest probability, a comparison with contrastive explanation methods should be included to ensure a more inclusive and in-depth evaluation.

3.

Concerning interpretability for human users, it's challenging to distinguish a substantial improvement of LIPEx over LIME, particularly in the sole visual example provided - Figure 8. Moreover, since the authors claim that their proposed method can effectively explain other classes, it would be beneficial to see more examples where LIPEx offers users a better overall understanding of model behaviors by checking the important features for different predicted classes.

Overall, I do not see a significant improvement of LIPEx over LIME and other current XAI methods at this moment.

**Questions:**

See weaknesses.

**Details Of Ethics Concerns:**

Not applicable.

---

### Official Review · Reviewer_pXhu · 2023-11-01

**Soundness:** 4 excellent
**Presentation:** 3 good
**Contribution:** 2 fair
**Rating:** 5
**Confidence:** 3

**Summary:**

This paper presents a new explainability framework building upon LIME, a method that uses simple, interpretable models to approximate and explain complex "black box" algorithms locally. The introduced framework, named LIPEx, specifically enhances this approach for multi-class classifiers by suggesting that valuable insights can be gained not just from the true class but also from examining the features associated with other classes.

For instance, in sentiment analysis, while understanding the features that contribute to a 'joyful' classification is important, analyzing features that are inversely related to a 'sad' classification can also provide meaningful explanations.

LIPEx achieves this by performing a linear approximation of the classifier's network up to the logits level; before the softmax operation is applied. It constructs a linear surrogate model based on perturbations of the input, which allows for the examination of feature associations for every class of interest through the model's weights. By extending the focus beyond the true class label, LIPEx offers more comprehensive explainability insights than its predecessor, LIME, which only considers the true class.

The idea is natural and the contribution is rather straightforward. The framework seems useful; however, there is room for improvement; especially regarding the presentation of the paper. I would be willing to increase my score if some of my questions are answered.

**Strengths:**

1) The idea is very intuitive and natural and the experimental results are thorough and convincing.
2) The intuition behind linear approximation being accurate for small perturbations in ReLU networks is very interesting.

**Weaknesses:**

1) Some issues with the figures and formatting: (i) The font sizes for Figures 2, 4, and 6 are not readable. (ii) Figures 10 onward do not have any captions and some of them are out of the page margin.
2) More convincing image examples are required. One where the improvement of LIPEx over LIME is evident.
3) LIPEx works only with neural networks that define logits and softmax layers; however, in general, the surrogate model framework aims to approximate the black box model without any knowledge about its inner workings. Therefore, LIPEx is somewhere between being a black box explainability method and an explainability method that knows the inductive biases of the architecture.

**Questions:**

1) Is there any ablation for using the Hellinger distance measure over other discrepancy methods such as KL divergence or total variation? Remember that just because the loss function is less for one does not mean it is better in explanations.
2) Why is $\pi(\mathbf{1}_s, x)$ defined this way? Is this standard? This means that for further away perturbation the Hellinger distance association should be more pronounced. But this goes against the fact that perturbations should be close to the original input.
3) Can you provide a similar table to Table 3 for the LIME baseline as well?
4) Why are the models in Algorithm 1 (in Appendix A) only considered to have linear least squares loss with $\ell_2$-norm?
5) Why do the TVLoss and HDLoss oscillate so much in Figure 5 (Appendix D.2)?

---

### Official Review · Reviewer_KfmR · 2023-11-03

**Soundness:** 2 fair
**Presentation:** 2 fair
**Contribution:** 2 fair
**Rating:** 3
**Confidence:** 4

**Summary:**

The paper proposes another method, named 'LIPEx', to explain individual predictions of black-box models. Its operations are similar to LIME: First, LIPEx creates a set of perturbed samples and obtains the corresponding predictions from the black-box model. Then, it uses the perturbed data and the predictions to train a surrogate model.

The proposed surrogate model remaps the black-box model's predictions into a matrix where one of the dimension represents the output classes and the other represents the features. This matrix is defined to be the "explanation matrix". The loss function used to learn the surrogate model has been modified in two ways: First, the weighing function has been changed to a custom one. Second, the distance metric has been changed to use the Hellinger distance.

The paper claims that LIPEx "provides insight into how every feature deemed to be important affects the prediction probability for each of the possible classes", is more efficient than LIME, and "causes more change in predictions for the underlying model than (other models)".

**Strengths:**

The use of an "explainable matrix" allows one to quickly visualize the importance of the features on the output classes, which is quite useful. It is also commendable that the paper has performed many test to compare its methods against other explainable A.I. methods and is transparent with the results. While the performance of LIPEx is only marginally better than the next best model, the good performance of LIPEx when a low number of perturbed data points is used can be an useful characteristic.

**Weaknesses:**

The presentation of the paper requires major revisions. This version is poorly organized, full of mistakes and confusing. However, I think that it can be a decent paper if the authors can write well.
1. The introduction uses many materials from the latter sections and can be shortened significantly.
2. Abbreviations should be defined at first mention (e.g., 'XAI' in the abstract').
3. The authors should describe the experiments in detail in the main text instead of explaining it in the captions (e.g., Table 3).
4. The authors can consider writing simpler and shorter sentences instead of long ones. Please pay attention to the grammar and your usage of commas.
5. Some equations can be defined in a clearer way and explained better in the text.

The authors can consider describing the framework laid out by LIME before clearly explaining how LIPEx modifies each component (e.g., weighing function, distance metric). The paper reads as though LIPEx is fundamentally different from LIME but I see many similarities.

The tests reveal that the performance of LIPEx is not too different from LIME on many datasets. The paper can benefit from a clearer emphasis of the advantages of LIPEx.

**Questions:**

Are you referring to perturbed data points when you use the term 'perturbations'? Please clarify your terminology.

How does the computational time of deploying LIPEx compare with LIME? Estimating a linear model is is quick, which is probably why LIME is so commonly used despite other proposed improvements.

Can the input data, defined as 'x' in your equations, take on negative values? If not, then aren't you just using a static weight in the loss function?

---

### Official Review · Reviewer_LRFa · 2023-11-04

**Soundness:** 1 poor
**Presentation:** 1 poor
**Contribution:** 2 fair
**Rating:** 3
**Confidence:** 4

**Summary:**

This paper introduces LIPEx, an XAI method which replicates the class-wise prediction probabilities of a complex model, and provides locally linear explanations via linear regression over data perturbations, with respect to the squared Hellinger distance. The authors study the sensitivity, faithfulness and stability of LIPEx, via distortions, perturbations and noise applied to the model and input data. They further compare LIPEx to LIME and existing gradient and path based saliency attribution methods, and present results for BERT on 20NewsGroups & Emotion; for VGG-16 and InceptionNetV3 on Imagenette.

**Strengths:**

1. The literature review is thorough and demonstrates a comprehensive understanding of the motivations, principles and existing methods in XAI.
2. The authors analyse the faithfulness (to the original model) and trustworthiness (robustness of explanations) via different experiments, centred around perturbing either the input data or the original model.
3. XAI for model debugging is an interesting and important task to work on in both image and text modalities.

**Weaknesses:**

1. **Unreliable Results.**  I have serious concerns about the reliability of numerical results in this paper:
- Top-K results are inconsistent between Tables 2 and 6 (despite representing the same set of image experiments). To elaborate, Table 2 presents numbers for Top-K, K=2, 3, 4; whereas Table 6 presents the exact same numbers but for K=3, 4, 5. For instance, Table 2 reports LIPEx Top-4 performance for VGG-16 as $0.867$ whereas this becomes $0.82$ in Table 7. Similar shifts are observed for all other rows and columns.
- Numbers are presented to arbitrary precision in Tables 1 & 2 (main text) and 6 & 7 (supplement): some values are given accurate to 1 decimal point (d.p.), some to 2 d.p., others to 3 d.p.
- Other more minor presentation issues include mislinking of Tables 6 and 7 (LaTeX references are swapped), misspelling of "Imagenette" as "Imagenetee".
- The numerical results are wildly inconsistent and should be thoroughly verified before any resubmission.

2. **Poor presentation.** The mathematical notation in this submission is unnecessarily convoluted; the manuscript contains both redundant definitions and missing definitions. It is confusing to follow and hard to understand. For instance, it takes one several rereadings to comprehend what LIPEx actually takes as an input; what the term "feature" refers to at different points of the paper.

3. **Limited evaluation.** LIPEx is evaluated on relatively simple datasets (20NewsGroups, Emotion and Imagenette) and architectures (BERT, VGG-16, InceptionNetV3). It is unclear how such local interpretability results translate to larger complex datasets WikiText-103 or ImageNet-1K, and various recent text/vision models.

4. **Non-standard image comparisons.** In Table 2, "top features detected by LIPEx" refer to "image segments which were obtained from Segment Anything", as clarified in the caption. LIME and LIPEx can directly leverage off-the-shelf SAM segmentation maps (only needing to "get a weight for each segment") whereas saliency-map based methods do not have direct access to these ground-truth segmentations (they need to first compute pixel-wise saliencies, which are then summed for each segmentation "feature" mask to constitute the feature "weight"). This is not a fair and standard comparison in vision XAI, since LIME and LIPEx only assign weights to precomputed segments whereas other methods are required to perform pixel-wise localisation.

4. **Limited novelty.** The LIPEx method tries to replicate the complex model's predictions with respect to the set of chosen features, with a locally linear model. It makes use of the squared Hellinger distance and $l2$ regularization, which are well-known and standard functions in existing literature.

**Questions:**

1. Could the authors kindly explain the inconsistent presentation of numerical results?
2. Could the authors address my concerns (as detailed in W3 & 4) regarding various limitations of image and text modality evaluations?